# Mechanism of assembly, activation and lysine selection by the SIN3B histone deacetylase complex

Mandy S. M. Wan [1,3], Reyhan Muhammad[1,3], Marios G. Koliopoulos[1], Theodoros I. Roumeliotis [2], Jyoti S. Choudhary [2] & Claudio Alfieri [1] ✉

Lysine acetylation in histone tails is a key post-translational modification that controls transcription activation. Histone deacetylase complexes remove histone acetylation, thereby repressing transcription and regulating the transcriptional output of each gene. Although these complexes are drug targets and crucial regulators of organismal physiology, their structure and mechanisms of action are largely unclear. Here, we present the structure of a complete human SIN3B histone deacetylase holo-complex with and without a substrate mimic. Remarkably, SIN3B encircles the deacetylase and contacts its allosteric basic patch thereby stimulating catalysis. A SIN3B loop inserts into the catalytic tunnel, rearranges to accommodate the acetyl-lysine moiety, and stabilises the substrate for specific deacetylation, which is guided by a substrate receptor subunit. Our findings provide a model of specificity for a main transcriptional regulator conserved from yeast to human and a resource of protein-protein interactions for future drug designs.

Histone acetylation activates transcription by inhibiting nucleosome-DNA interactions thereby promoting chromatin decompaction, and by recruiting transcriptional coactivators that recognise this modification[1–3]. This histone mark is "erased" by the activity of histone deacetylases (HDACs). Structurally, HDACs feature a conserved α/β deacetylase fold with a central β sheet surrounded by α helices at both the sides[4]. Class I HDACs (HDAC1/2/3/8) represent the principle nuclear deacetylases that regulate transcription. HDAC1 and HDAC2 (HDAC1/2) are essential for cellular differentiation, development[5–8] and function as part of multi-subunit complexes[9]. HDAC1/2 are $Zn^{2+}$-dependent enzymes which catalyse the nucleophilic attack of a $Zn^{2+}$-bound water molecule to a $Zn^{2+}$-polarised N-ε-acetyl-lysine side-chain carbonyl. This generates a tetrahedral intermediate that collapses by protonation from a catalytic histidine residue, which generates acetate and lysine[10]. Besides the catalytic zinc atom, class I HDACs interact with two additional cations, that are required for activity. In the human HDAC2 crystal structure two $Ca^{2+}$ are found[11]. The first one directly binds an aspartate residue, which lowers the $p$Ka of the catalytic

histidine, and the second one binds on a second binding site ~21 Å away from the active site at the opposite side of the HDAC central β sheet[10]. Intriguingly, catalytic activity and substrate/gene specificity of HDAC1/2 in isolation is very poor or absent respectively[9]. These physiological roles emerge when HDAC1/2 form specific complexes with corepressor proteins such as the SIN3 proteins[12–15]. How substrate specificity and catalytic activation is achieved in these holo-complexes is incompletely understood since we are lacking biochemical and structural information on the intact version of these complexes, which are challenging to reconstitute and structurally analyse because of their high intrinsic disorder.

In the context of NuRD[16], MiDAC[14,17], and SMRT[18] HDAC complexes, a SANT domain from a corepressor subunit stimulates the HDAC1/2 activity via binding of an inositol phosphate (InsP) which functions as a molecular "glue" between the SANT domain and the enzyme. Subsequent studies have shown that InsP contacts an allosteric basic patch formed by the HDAC α1 and α2, which influences residues in its catalytic site[19]. Conversely, it is not clear if SIN3

[1]Division of Structural Biology, Chester Beatty Laboratories, The Institute of Cancer Research, London, UK. [2]Functional Proteomics, Chester Beatty Laboratories, Cancer Biology Division, The Institute of Cancer Research, London, UK. [3]These authors contributed equally: Mandy S. M. Wan, Reyhan Muhammad. ✉e-mail: claudio.alfieri@icr.ac.uk

complexes are regulated by InsP and the overall multi-subunit assembly of these complexes is also unknown.

In mammals, two highly similar proteins named SIN3A and SIN3B[20], are part of two distinct mammalian SIN3 complexes[21,22]. These two complexes are homologous to the *Saccharomyces cerevisiae* Rpd3(L) and Rpd3(S) complexes respectively[21,23–26]. Depletion of SIN3A causes loss of proliferative potential[27–29] and depletion of SIN3B inhibits the ability of proliferating cells to exit the cell cycle[30–32]. Because of the ability of SIN3 complexes to regulate cell differentiation[5–7] and proliferation in cancer cells, they are drug targets[33–36]. Consequently, several inhibitors have been developed and used in clinical trials[37,38], however success on this has been limited by the lack of specificity of these compounds towards the distinct HDAC1/2 complexes[17].

Understanding the assembly of different subunits, within the holo-SIN3 complexes is therefore critical to understand the mechanisms of substrate selection and activation in vivo and to design more specific and potent inhibitors for anti-cancer therapy. Here we report the cryo-electron microscopy (cryo-EM) structure of the human SIN3B complex in apo form and bound to suberoylanilide hydroxamic acid (SAHA)[17,37], which mimics an acetyl-lysine substrate. Our structural data combined with complementary cross-linking mass spectrometry (XL-MS) and biochemical assays show the molecular basis of complex assembly, complex-dependent stimulation of catalytic activity, and lysine selection in the context of an acetylated nucleosome substrate.

## Results

### Reconstitution of recombinant human SIN3B complex
It has been proposed that the human SIN3B complex consists of the SIN3 scaffold subunit, one catalytic core (i.e. HDAC1/2), and two chromatin targeting modules: the plant homeodomain (PHD) containing protein PHF12, and the MRG (MORF4-related gene) and chromo domain-containing protein named MORF4L1 (mortality factor 4-like 1)[21,39]. However, clear evidence of direct protein:protein interaction among these subunits has been lacking. In order to address this, we reconstituted a recombinant version of the human SIN3B complex, composed of the subunits SIN3B isoform 2, HDAC2 (hereafter named HDAC for simplicity), PHF12 and MORF4L1 (Fig. 1a), expressed in the baculovirus/insect cell system. The resulting complex is stable in size-exclusion chromatography, stoichiometric, highly pure, and enzymatically active as it manifests HDAC activity with a fluorogenic acetyl-lysine substrate (see "Methods", Supplementary Fig. 1a, e).

### Cryo-EM structure of the SIN3B complex
In order to characterise the stoichiometry and architecture of the SIN3B complex, as well as the molecular mechanism of HDAC activation by SIN3B, we pursued the structure determination of this complex by cryo-EM (see "Methods", Supplementary Fig. 2 and Supplementary Table 1). 2D class-averages of this complex show V-shaped particles of ~150 Å size (Supplementary Fig. 2a, b). Ab initio reconstruction of this complex followed by 3D-refinements and classifications allowed us to obtain a high-resolution map of the full-length SIN3B complex at 3.4 Å resolution (Supplementary Figs. 2c and 5). Moreover, we obtained an additional map of the catalytic SIN3B core subcomplex (SIN3B^core) at the resolution of 2.8 Å (Supplementary Fig. 3 and Supplementary Movie 1). The SIN3B^core encompasses most of the SIN3B protein, including its third paired amphipathic α-helix (PAH3) domain, a previously unknown middle-domain (MD), the HDAC interacting domain (HID) domain, and a SIN3B C-terminal domain (CTD), which was previously defined[12] as PAH4 and highly conserved region (HCR) (Fig. 1). Strikingly, SIN3B wraps around the catalytic HDAC module with its MD, HID, and one loop coming from the CTD (Fig. 1c–e). Moreover, a loop (i.e. Gate loop) connecting MD and HID inserts into the active site of the HDAC (Fig. 1c, d). SIN3B^core also includes PHF12, which mediates interactions with all the subunits of SIN3B. PHF12 includes one previously uncharacterised N-terminal helix (NTH), one PHD domain

(PHD2), and one previously predicted SIN3 interacting domain[40] (SID2) (Fig. 1a, c). In the SIN3B complex, an additional histone recognition module sits on top of SIN3B^MD and HDAC, and includes the MRG domain of MORF4L1, which interacts with the PHF12^SID1, and the first PHD domain of PHF12 (PHD1) (Fig. 1c and Supplementary Movie 1).

In conclusion, SIN3B, HDAC, PHF12 and MORF4L1 interact with a 1:1:1:1 stoichiometry to form a V-shaped complex where the SIN3B^CTD and substrate recognition module form two arms at the top, the SIN3B^HID at the bottom, the HDAC is in the centre, and the SIN3B^PAH3 connects the two arms in such a way that the HDAC is completely encircled by the SIN3B subunits (Fig. 1c, f and Supplementary Movie 1).

### SIN3B establishes multiple interactions with HDAC
The SIN3B interaction with HDAC is extensive and buries about one third of the HDAC surface area (i.e. 3566 Å²) (Supplementary Movie 1 and Fig. 1d, e). The first remarkable interaction between SIN3B and HDAC involves the SIN3B^MD, a short domain containing a beta-sheet and one short helix (α-13). The beta-sheet forms by the contribution of two beta-strands (i.e. β1 and β2) N-terminal to α-13 and one strand (i.e. β3) C-terminal of α-13 (Fig. 2a and Supplementary Fig. 6). SIN3B^MD packs against the second calcium (Ca2) binding site of HDAC, which has been shown to be important for the activity of class I HDACs (Fig. 2a)[4,10]. This interaction features mainly hydrophobic contacts involving SIN3B^MD beta sheet Tyr401 with HDAC Tyr189, and SIN3B α-13 Leu424 with HDAC Leu162 and Phe188. Moreover, SIN3B Asn425 and HDAC Glu186 are involved in side-chain-backbone interactions which stabilise the MD:HDAC interaction (Fig. 2a). Consistently, our XL-MS data on the SIN3B complex show several cross-links between the MD and residues near HDAC Tyr223, the backbone of which interacts with the Ca2 atom (Supplementary Fig. 7a and Supplementary Data 1). In SIN3B isoform 1, Leu424 (Fig. 2a and Supplementary Fig. 6a) is replaced by one aspartate residue, which would interfere with the hydrophobic contacts between MD and HDAC. This provides a structural explanation to the results from previous HDAC activity assays showing that SIN3B isoform 2 is more active than the isoform 1[39].

In conclusion, SIN3B interacts extensively with HDAC, among these interactions the SIN3B^MD contacts the Ca2 binding site of HDAC.

### The SIN3B^HID directly contacts the HDAC basic patch for catalytic activation
As already mentioned, the NuRD[16], MiDAC[14,17], and SMRT[18] HDAC complexes are activated by InsP, which interacts with both the HDAC allosteric basic patch and a SANT domain belonging to a corepressor subunit[18,19].

Strikingly, in our structure the SIN3B HID domain binds directly the allosteric basic patch of HDAC, preventing InsP binding (Fig. 2b, c). This interaction is dominated by a triad of SIN3B acidic residues coming from the stalk region of the HID i.e., Glu456, Asp457 and Glu461 (Fig. 2c and Supplementary Fig. 6a). Specifically, Glu456 and Asp457 interact with HDAC Lys32 and Arg271 via a network of hydrogen bonds involving water molecules. The latter residues are involved in stimulating HDAC activity in HDAC complexes regulated by InsP[16,18,41]. Moreover, Glu456 interact with the backbone of the catalytic Tyr304, which as illustrated below is involved in substrate binding. These interactions are consistent with cross-links we found between SIN3B HID and the C-terminal portion of HDAC2 (Fig. 2c and Supplementary Fig. 7a). The SIN3B^HID acidic triad occupies a similar position as the InsP in the structure of SMRT:HDAC complex[18] (Supplementary Fig. 8a). Consequently, in the SIN3B complex, the acidic triad of SIN3B^HID would compete with InsP binding. Therefore, it would be expected that the InsP would not further stimulate the catalytic activity of the SIN3B complex. We tested this hypothesis experimentally and found that although the HDAC activity in the SIN3B complex is higher than the one of the HDAC apo enzyme, there is no further stimulation

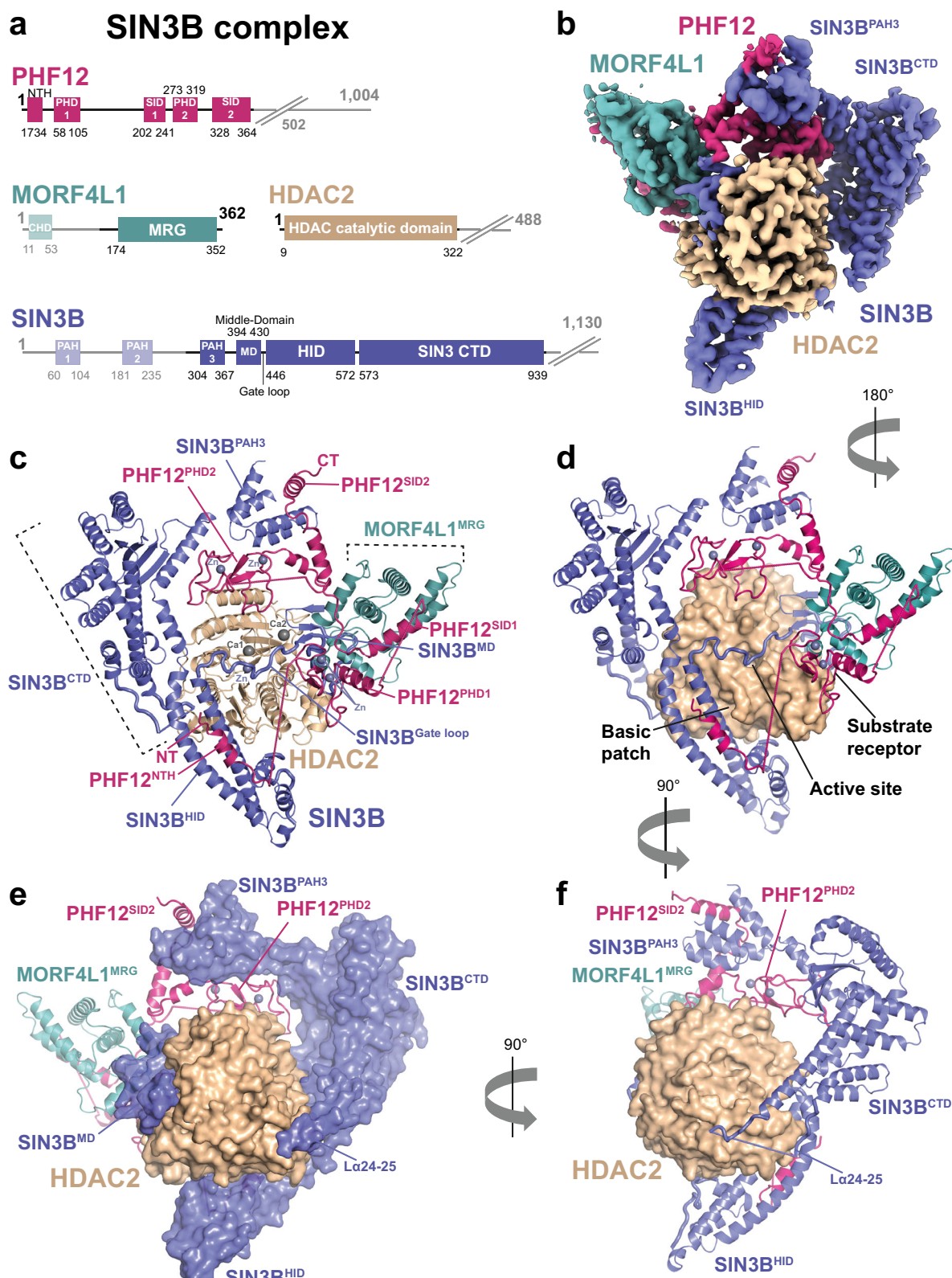

**Fig. 1 | Overall architecture of the SIN3B apo complex. a** Schematics showing domains and functional regions on the subunits of the SIN3B complex represented in linear form. SIN3B subunits are SIN3B (blue), HDAC2 (wheat), MORF4L1 (teal) and PHF12 (warm pink). Disordered parts of the complex are represented with fading colours. The PHF12$^{\Delta C\text{-terminus}}$ construct used in the SIN3B complex sample lacks the poorly conserved and unstructured C-terminus (as observed in our EM analysis) and it ends at residue 502. **b** Cryo-EM density of the SIN3B full-length (SIN3B$^{FL}$) sample at 3.4 Å resolution. **c**–**f** Cartoon representations of three main views of the SIN3B complex showing the overall architecture of the complex. Zinc (Zn) and calcium (Ca) cations are depicted and shown in shades of grey. HDAC2 is represented as surface in (**d**)–(**f**). SIN3B is represented as surface in (**e**).

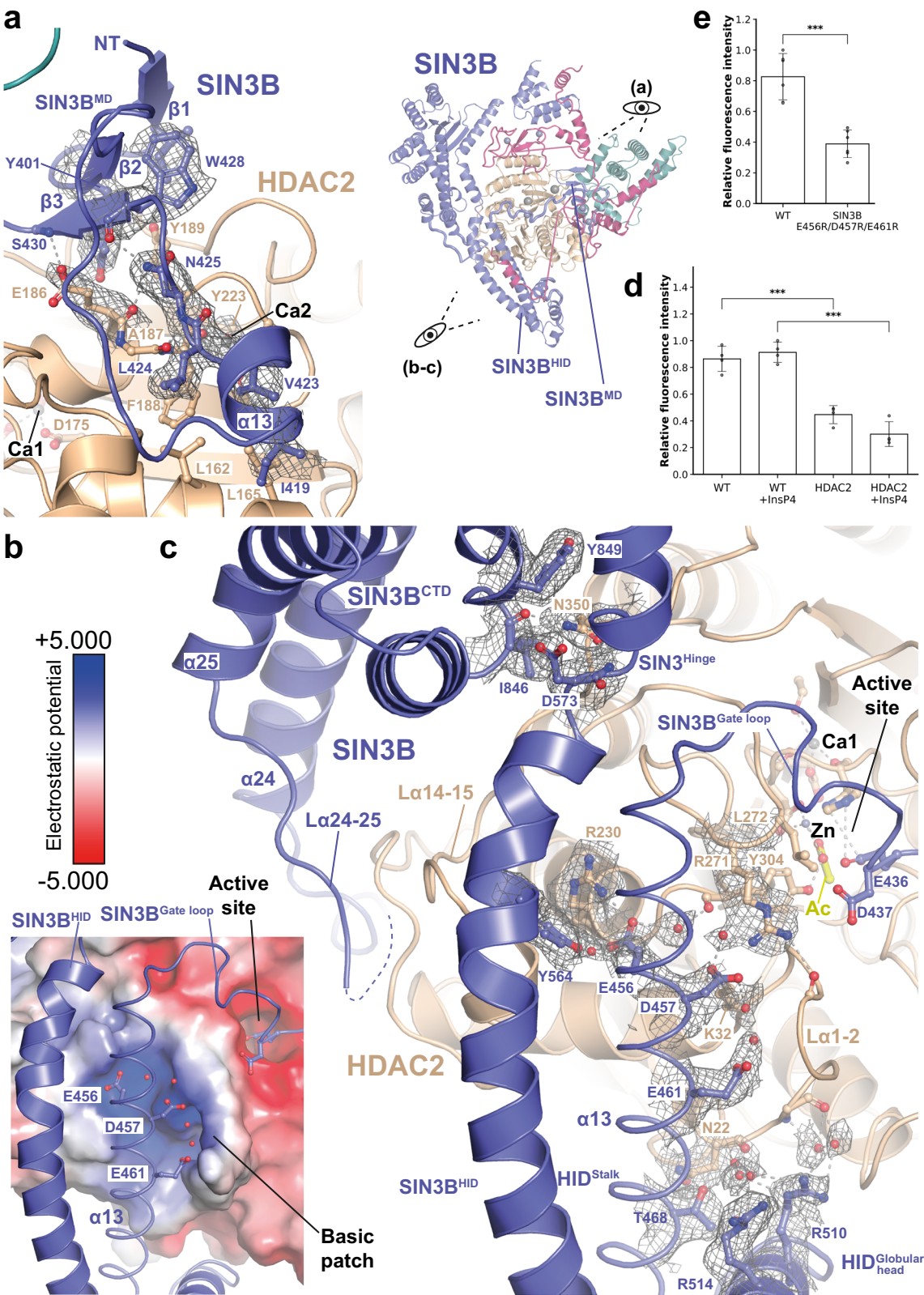

after addition of inositol-1,4,5,6-tetraphosphate (InsP4) (Fig. 2d). Consistently with reference[16] InsP4 does not stimulate the activity of the HDAC apo enzyme in the absence of a SANT domain containing corepressor binding partner (Fig. 2d). These results suggest that the SIN3B acidic triad functionally replaces the InsP cofactor in the SIN3B complex. In order to assess that this is the case, we generated a SIN3B complex where the acidic triad of SIN3B$^{HID}$ is replaced by arginine

residues. Indeed, we found that the HDAC activity of this charge-swap mutant is compromised (Fig. 2e).

The HID:HDAC interaction is also buttressed by the globular head of the HID that packs against the HDAC loop connecting α1 with α2 (L α1-2) (Fig. 2c). At the opposite side, the SIN3B hinge between HID and CTD form a pocket for HDAC loop α14-15 Asn350 (Fig. 2c). Further down, the loop α24-25 coming from the SIN3B$^{CTD}$ binds HDAC in such a

**Fig. 2 | HDAC activation by SIN3B. a–c** Interactions between SIN3B and HDAC. SIN3B$^{core}$ cryo-EM density map at 2.8 Å resolution is shown for relevant sidechains and water molecules. Colour scheme for SIN3B subunits as in Fig. 1. **a** Close-up view on the interaction between the SIN3B middle domain (SIN3B$^{MD}$) and the HDAC second calcium ion (Ca2) binding site is shown on the left. Overall structure of the SIN3B complex and relation to zoomed view shown in this figure is illustrated on the right. Residues involved in hydrophobic and backbone-mediated interactions are depicted. **b, c** Close-up view on the interaction between SIN3B HDAC interacting domain (HID) and the HDAC basic patch, which in other class I HDAC complex is involved in activation by Inositol phosphates (InsPs). The electrostatic surface potential (±5.000) is shown on HDAC in (**b**). SIN3B$^{HID}$ residues involved in interactions with HDAC2 are depicted. Most of these interactions are water mediated. **d, e** HDAC activity assay performed with either HDAC alone, SIN3B complex and, SIN3B complex point mutants. Data are presented as mean values ± S.D. of independent experiments, $n = 4$ for (**d**) and $n = 6$ for (**e**). ***$p < 0.0005$ (one tailed unpaired $t$-test). In (**d**) $p = 0.181e^{-05}$ (WT vs. HDAC2), $p = 2.504e^{-05}$ (WT + InsP4 vs. HDAC2 + InsP4). In (**e**) $p = 5.837e^{-05}$ (WT vs. E456R/D457R/E461R).

way that this region and the SIN3B$^{MD}$ hug completely HDAC from opposite sides (Figs. 1e and 2c).

In conclusion, our structural data show that SIN3B contacts HDAC in several key surfaces thereby stimulating HDAC activity and that this stimulation is independent from the InsP4 co-factor but relies on direct protein-protein interactions established by the SIN3B corepressor. This represents an unanticipated mechanism of Class I HDAC activation by corepressor proteins.

### The SIN3B$^{Gate loop}$ inserts into the HDAC catalytic tunnel

As detailed above, our cryo-EM structure of the SIN3B complex shows that a loop connecting MD and HID domains (Gate loop) inserts into the active site of the enzyme (Fig. 2c, Supplementary Fig. 6a and Supplementary Movie 1). SIN3B Glu436 enters the active site and its carboxyl group establishes hydrogen bonds with the catalytic HDAC His179, whereas the Cβ and Cγ of Glu436 makes non-polar contacts with HDAC Phe206 (Fig. 3a, c). The Cβ of the following SIN3B Asp437 makes non-polar contacts with the HDAC hydrophobic Leu272 and Pro30. Similarly to ref. 16, we could also find an acetate molecule coordinating with the catalytic zinc cation and the catalytic Tyr304 (Fig. 3c). This suggests that our SIN3B apo structure represents a post-hydrolysis state of the deacetylation reaction where the protein substrate has been released and the gate loop has returned to a conformation in which it occludes the active site.

### Acetyl-lysine recognition by the SIN3B complex and regulation by the SIN3B$^{Gate loop}$

In order to gain insights into the acetyl-lysine substrate recognition by the SIN3B complex and into the functional role of the SIN3B$^{Gate loop}$, we determined the cryo-EM structure of the SIN3B complex with the inhibitor SAHA, which mimics a substrate acetyl-lysine and the peptide backbone of the substrate +1 residue[17,37] (Supplementary Fig. 4, Supplementary Table 1 and Supplementary Movie 1). Strikingly, the SIN3B$^{Gate loop}$, which in SIN3B apo structure occludes the catalytic site, flips 180 degrees, thereby providing access for SAHA binding in the catalytic tunnel (Fig. 3b, d). SAHA replaces SIN3B residues Glu436 and Asp437 and the acetate molecule, which were present in the catalytic site of the SIN3B-apo structure (Fig. 3a, c), and its binding mode is comparable with the previous crystal structure of the same drug with HDAC[42] where the hydroxamic acid moiety coordinates the HDAC zinc cation, establishes hydrogen bonds with His141-142 and the catalytic Tyr304, and the aliphatic chain participates in hydrophobic contacts with HDAC Phe206 and stacks with HDAC His179 (Fig. 3d). The amide NH group connecting the aliphatic chain and the phenyl ring performs a hydrogen bond with HDAC Asp100, thereby giving specificity to this interaction by mimicking the substrate backbone +1 residue (Fig. 3d).

Strikingly, in our structure, the phenyl ring of SAHA which mimics a substrate +1 residue, stacks between the SIN3B$^{Gate loop}$ Phe440 and the HDAC His29 and Pro30, thereby suggesting that the SIN3B$^{Gate}$$^{loop}$ participates in stabilising the histone substrate in the active site. In order to experimentally assess the functional importance of the SIN3B$^{Gate loop}$ in deacetylation of a genuine histone substrate we tested the ability of SIN3B to deacetylate a nucleosome carrying the histone H3 lysine 27 acetylation (H3K27ac) (Fig. 3e, f). Strikingly, mutating the conserved Glu436 and Asp437 to alanine (i.e. SIN3B$^{E436A/D437A}$ mutant,

Supplementary Figs. 1c and 6a), which is predicted to displace the SIN3B$^{Gate loop}$ away from the HDAC active site, abolishes the ability of SIN3B to deacetylate H3K27 from a nucleosome (Fig. 3e, f). Our results show that the SIN3B$^{Gate loop}$ should be in proximity of the active site for establishing interactions with the histone tail substrate and ensure substrate-specific deacetylation. Notably, although the SIN3B$^{E436A/D437A}$ mutant is no longer able to deacetylate the specific H3K27 from a nucleosome, it still manifests HDAC activity with the fluorogenic acetyl-lysine substrate (Fig. 3g). These data together support the idea that the SIN3B$^{Gate loop}$ allows for deacetylation of the specific target substrate (i.e. the acetylated nucleosome), where the lysine residues within the long histone tail need to be optimally presented into the SIN3B complex active site. Conversely, the fluorogenic substrate used in our HDAC assay, given its small size, could have easier access to the SIN3B complex active site, where it can still be processed in the SIN3B$^{E436A/D437A}$ mutant.

In conclusion, we find that histone deacetylation by the SIN3B complex involves the joint contribution of the HDAC active site residues and the SIN3B$^{Gate loop}$, which together position the histone substrate for deacetylation.

### PHF12 stabilises the SIN3B:HDAC complex and forms a histone recognition module

Our structure shows that PHF12 is key in defining the SIN3B architecture since it contacts each core subunit domain and it also forms a histone tail recognition module (Figs. 1 and 4). Firstly, PHF12$^{NTH}$ binds the stalk region of the HID in a way which is reminiscent of the SIN3A$^{HID}$:SDS3$^{SID}$ interaction[43] (Fig. 4a and Supplementary Fig. 8b). PHF12$^{NTH}$ and SDS3$^{SID}$ bind SIN3$^{HID}$ in a mutually exclusive manner as the N-terminal extended portion of the SDS3$^{SID}$ would clash with the C-terminal portion of the PHF$^{NTH}$, which is in an extended conformation (Fig. 1 and Supplementary Fig. 8b). The PHF12$^{NTH}$:SIN3B$^{HID}$ interaction is also supported by our XL-MS data (Supplementary Fig. 7a).

Strikingly, PHF12$^{PHD1}$ recognises a composite interface formed by the MORF4L1 MRG domain interacting with the PHF12$^{SID1}$ (Fig. 4b, c) as also supported by our XL-MS data (Supplementary Fig. 7a). Our structural data is agreeing with previous binding studies conducted by NMR showing that PHF12$^{PHD1}$ binds to MORF4L1$^{MRG}$ in a PHF12$^{SID1}$-dependent manner[44]. Furthermore, our structural data explains why the yeast MORF4L1 homologue (i.e. Eaf3) dissociates from the SIN3B complex when the SID1 of Rco1 (i.e. yeast PHF12) is deleted[45]. The MRG:SID1:PHD1 ternary assembly form the histone recognition module (or substrate receptor) of SIN3B. Within this structure the PHF12$^{PHD1}$ is located on top of the active site of the HDAC, suggesting that this domain could recruit the histone tail substrate to the enzyme (Fig. 4a, c). In fact, the PHF12$^{PHD1}$ has been shown to bind the unmodified end of histone H3[44].

Conversely, PHF12$^{PHD2}$ is an atypical PHD domain at the sequence level and from previous biochemical data it is not able to bind histone tails on its own[44]. Furthermore, in our structure this domain is wedged between the SIN3B$^{CTD}$ and the HDAC thereby stabilising the SIN3B:HDAC interaction (Fig. 4d). This interaction is mainly specified by the SIN3B α21 Glu602, which protrudes out from the SIN3B$^{CTD}$ and interacts with PHF12 Arg282. Moreover, HDAC Arg230 interacts with PHF12 Met298, Asp299 and Thr306 (Fig. 4d). Therefore, our structure

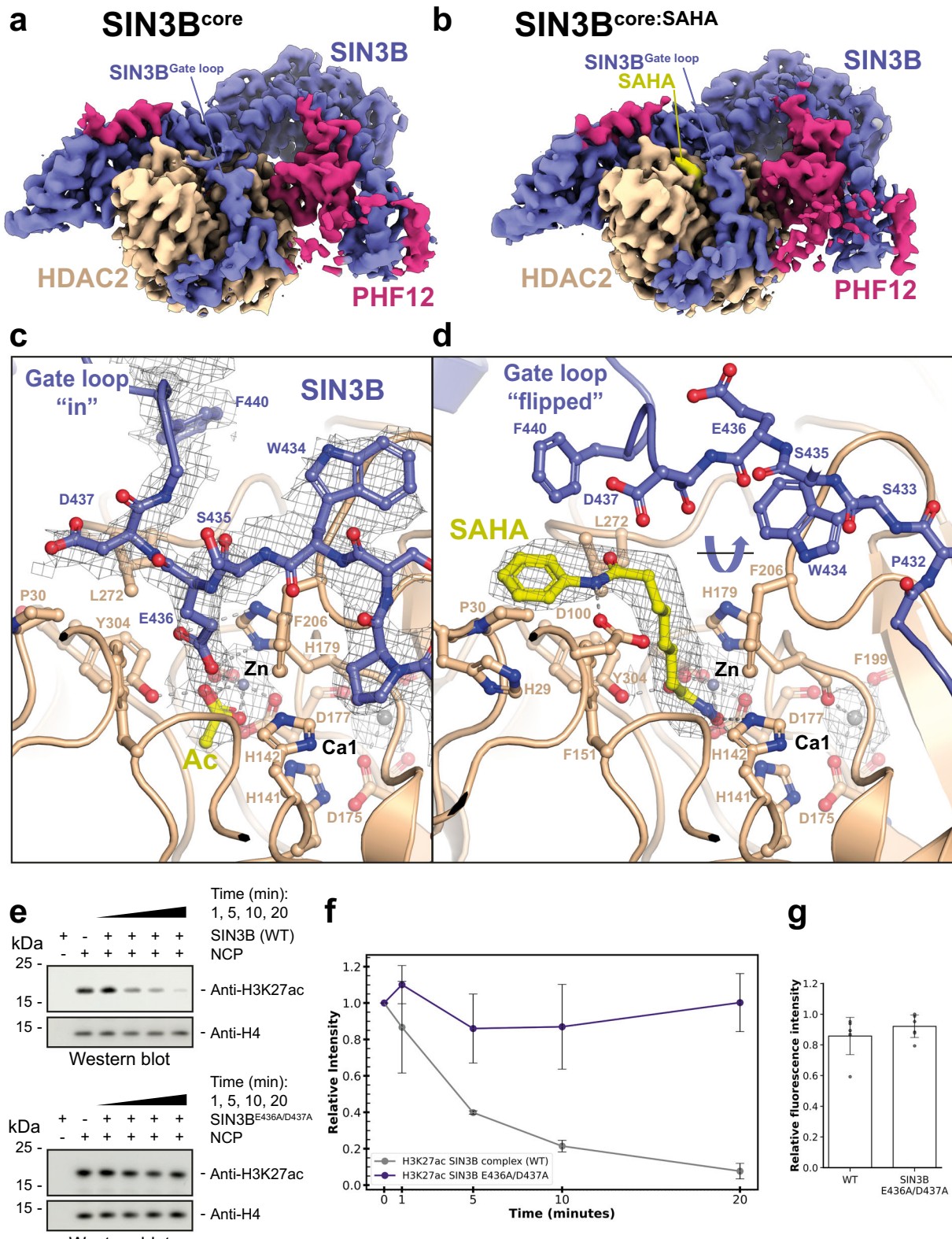

**Fig. 3 | Structure of the SIN3B complex with and without the acetyl-lysine mimic SAHA.** a, b Overall cryo-EM densities for SIN3B$^{core}$ and SIN3B$^{core}$:SAHA complex structures both at 2.8 Å resolution are shown. **c, d** Close-up view on the catalytic core of either the SIN3B$^{core}$ complex (**c**) or the SIN3B$^{core}$:SAHA complex cryo-EM structure determined here (**d**). The SIN3B$^{Gate loop}$, which is displaced by SAHA binding is highlighted. Acetate (Ac) and SAHA molecules are depicted in yellow. Zinc cations are shown in grey. Other subunits of the complexes are coloured as in Fig. 1. **e, f** Deacetylation reactions by the SIN3B complex of

nucleosomes with the acetyl moiety localised in lysine 27 (i.e. H3K27ac). The same experiment is also performed with a SIN3B Gate loop mutant (SIN3B$^{E436A/D437A}$) complex. Quantifications from western blot data (one replicate is shown in **e**) are plotted in (**f**). Data are presented as mean values ± S.D. of independent experiments, $n = 3$. **g** HDAC activity assay performed with either SIN3B complex and SIN3B complex point mutant. Data are presented as mean values ± S.D. of independent experiments, $n = 7$.

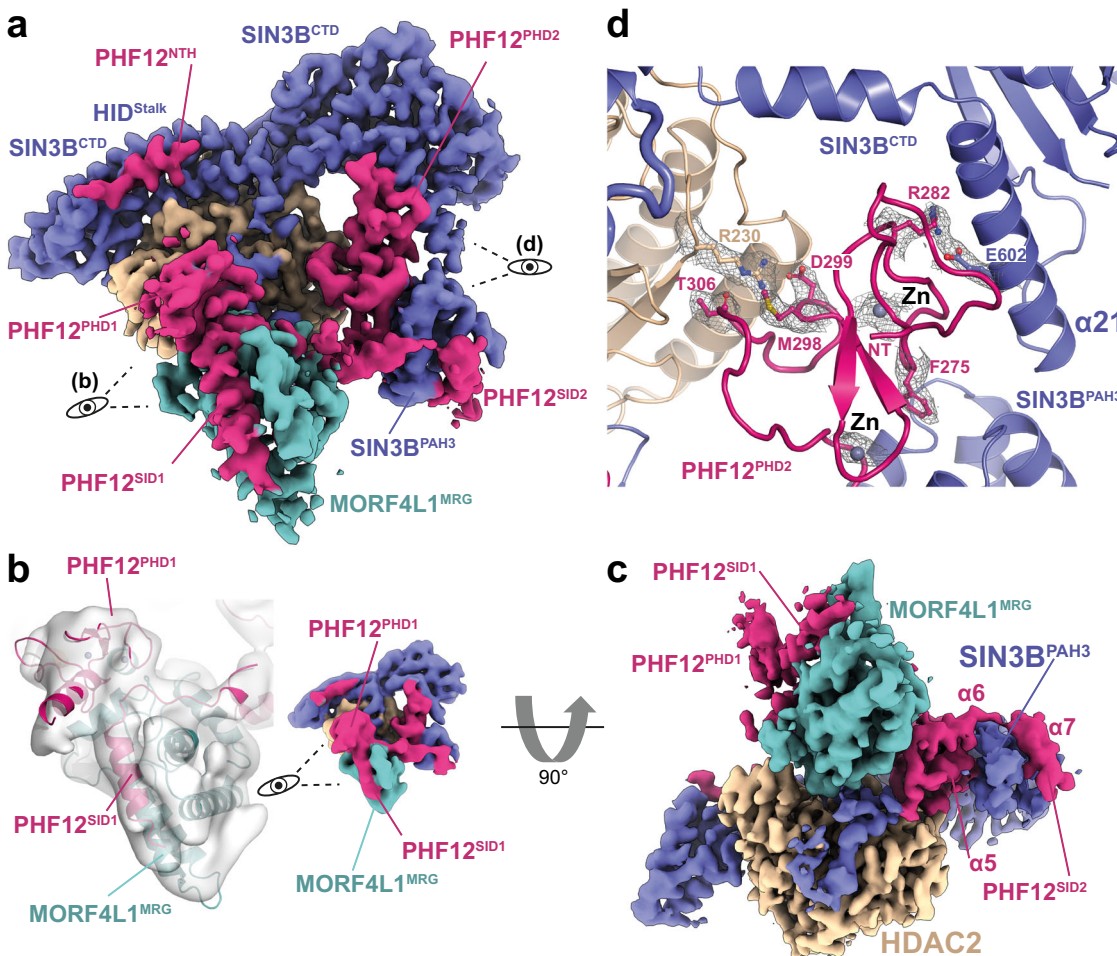

**Fig. 4 | Substrate recognition by SIN3B. a** Overview of the SIN3B^FL complex cryo-EM density at 3.4 Å resolution. PHF12 (magenta) is highlighted here and its PHD finger domains are shown in close up view in (**b**, PHD1) or (**d**, PHD2). **b** Close-up view on the PHF12 interactions with MORF4L1 MRG domain. PHF12^PHD1 recognises a composite interface of MRG and the PHF12 SIN3B interaction domain 1 (SID1). Cryo-EM map of the SIN3B^FL reconstruction is here shown lowpass filtered to 7 Å for better showing the PHF12^PHD1 domain (left) and the overall lowpass filtered map is shown on the right. **c** The cryo-EM map from (**a**) is shown rotated 90 degrees to highlight the interactions of PHF12^SID2 with SIN3B^PAH3 and MORF4L1 MRG domain. **d** Close-up view on the PHF12^PHD2 interacting with SIN3B and HDAC2. Key residues involved in these interactions, zinc (Zn) atoms and their cryo-EM densities are depicted.

demonstrates that PHF12^PHD2 has a structural role rather than a substrate recruitment role. Consistently, modelling of a PHF12^PHD2:H3 complex manifests clashes of H3 with SIN3B and with a loop preceding PHF^PHD2 (PHF^pre-PHD2) (Supplementary Figs. 6b and 8c).

In conclusion, our data suggests that PHF12^PHD1 and not PHF12^PHD2 can deliver the histone H3 substrate to the HDAC active site.

## A second SID in PHF12 binds the SIN3B PAH3 domain

Our structure shows that a second SID C-terminal to the PHF12^PHD2 binds to the SIN3B^PAH3 (Fig. 4c). Importantly, this is consistent with our XL-MS data (Supplementary Fig. 7a). The PHF12^SID2 is reminiscent of the SIN3A^PAH3:SAP30 interaction[46] since both these SIDs are tri-partite and form an extensive interaction with the PAH domain (Supplementary Fig. 9). In our structure, the C-terminal α7 from PHF12 inserts into the PAH cleft formed by four amphipathic helices by means of hydrophobic interactions by the PHF12 helical motif VDFLNRIH (Supplementary Fig. 6b and Fig. 4c). More N-terminally, PHF12 α6 is loosely attached to the SIN3B^PAH3 contacting rather the MORF4L1^MRG (Fig. 4c). Conversely, PHF12 α5 together with the preceding PHF12^PHD2 packs against the PAH3 domain at the opposite side of the PHF12 helix 7, thereby stabilising the interaction of PAH3 with both PHF12 and α21 from SIN3B^CTD (Fig. 4c, d). Therefore, our structural data indicates that the PHF12^SID2 is another crucial component involved in SIN3B complex assembly and it is critical for firmly anchoring the histone recruiting protein PHF12 to SIN3B. Functional importance of the PHF^SID2 in targeting SIN3B to chromatin has been demonstrated in one study[47] showing that a C-terminal deletion including the SID2 region in the yeast PHF12 (i.e. Rco1) affects nucleosome binding by the SIN3B complex.

In conclusion, we find that PHF12 is a key scaffolding subunit of the SIN3B complex which contributes to complex assembly by contacting each core subunit domain, and likely constitutes a substrate receptor by recruiting the H3 histone tail with its PHD1. The structural importance of this subunit that we observe is consistent with the strong phenotypes observed in the PHF12 knockout[48]. We also find that PHF12 competes with the SIN3A-associated SDS3 and SAP30 therefore explaining why these proteins take part in the two distinct SIN3A and SIN3B complexes[26]. Nevertheless, to fully understand the specificity mechanisms that cause the assembly of SIN3A and SIN3B proteins into two different complexes, requires future endeavours in determining the high-resolution structure of the SIN3A complex.

## Specific SIN3B deacetylation of p300 histone marks on the histone H3

SIN3B contributes to gene repression by counteracting the activity of the HAT p300 at target promoters[49–56] (Fig. 5a, b). Our structure is

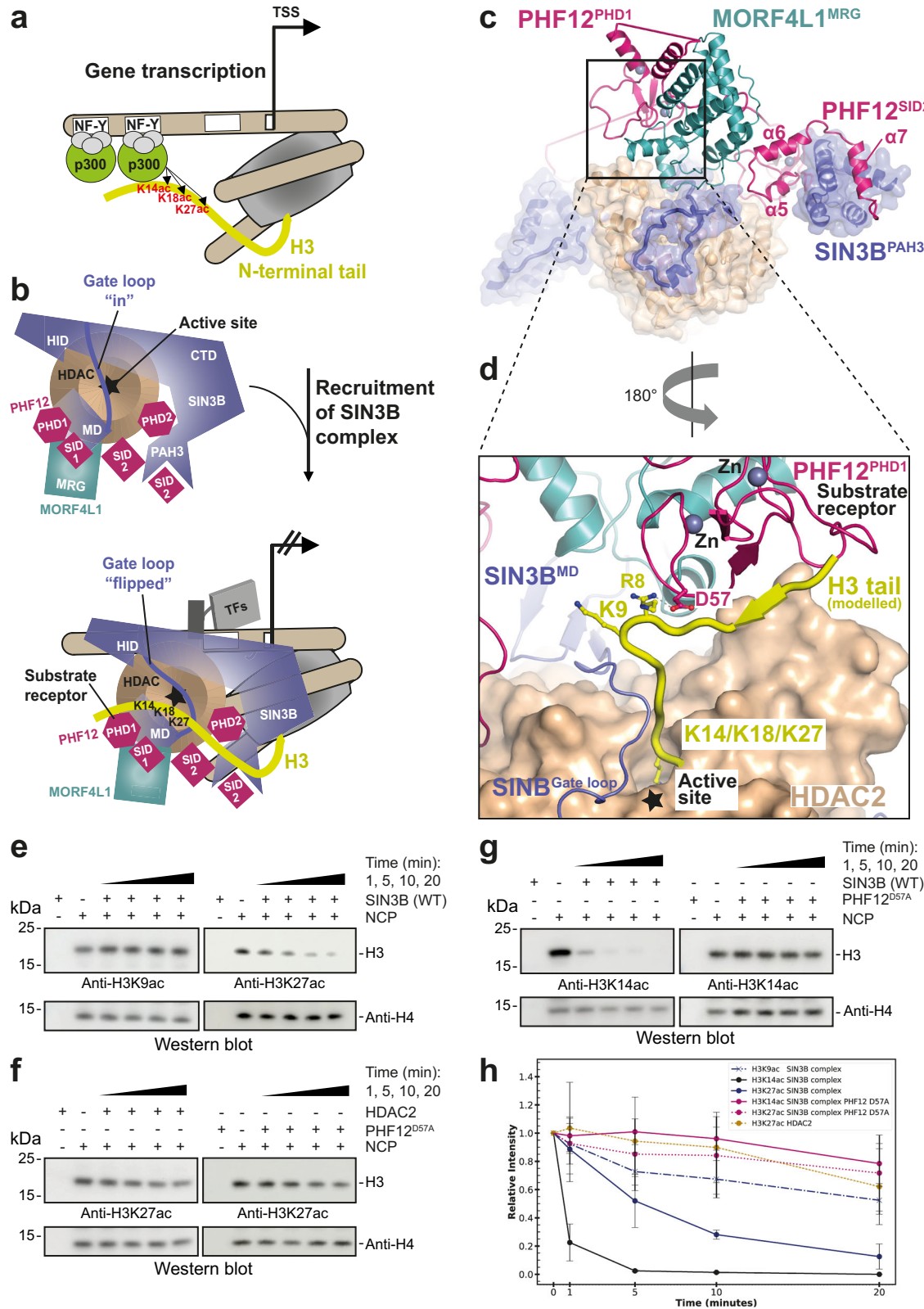

consistent with a specific role of the SIN3B complex in removing the p300 specific mark H3K27[52,57] instead of the more N-terminal acetylation H3K9 (Fig. 5c, d). In fact, molecular modelling of an H3 histone tail substrate:SIN3B complex based on (1) the acetyl-lysine position shown by our SIN3B:SAHA structure, and (2) the superposition of the crystal structure of the BHC80[PHD]:H3[58] with the PHD1 in our structure, indicates that PHF12 could present the histone H3 tail to the catalytic

tunnel of the HDAC starting from Lys14 (Fig. 5d). In order to demonstrate this, we probed deacetylation of SIN3B on nucleosomes containing either H3K9ac or H3K27ac marks. As result, SIN3B can deacetylate nucleosomes containing either H3K27ac (Fig. 5e) or H3K14ac (Fig. 5g), but is unable to deacetylate nucleosomes containing H3K9ac, which is not a p300-specific histone mark (Fig. 5e, h). Strikingly, deacetylation of H3K27 is compromised if using either the HDAC

**Fig. 5 | Model of SIN3B function at gene promoters. a, b** Schematic cartoon illustrating a model for the function of SIN3B complex at target genes. SIN3B counteract p300-mediated acetylation. p300 binds the NF-Y sequence at target genes, whereas p300 can be recruited by various transcription factors (TFs) and their DNA-specific sequence (indicated with a white box). The HDAC active site is indicated with a star. The conformational change of the SIN3B^{Gate loop} upon histone H3 tail binding is indicated. H3K27/K18/K14 (and not H3K9) are main acetylation targets by p300[52,56]. **c** overall structure of SIN3B complex is shown in cartoon and transparent surface representation. **d** Modelling of the H3 histone tail based on the structure of BHC80^{PHD}:H3 (PDB ID: 2PUY) superposed to our SIN3B structure (PHD1 as a reference) and based on our SIN3B:SAHA structure. Lysine residues on the H3 N-terminus are shown, K9 would be too far from the catalytic site, lysine residues more C-terminal to H3 residue 13 (i.e. K14, K18 and K27) would be able to enter the HDAC catalytic site. **e–h** Deacetylation reactions by the SIN3B complex of nucleosomes with the acetyl moiety localised in different positions along the H3 histone tail (i.e. H3K9ac, H3K14ac and H3K27ac). The H3K27/K14 deacetylation experiments are also performed with a SIN3B PHF12 PHD finger 1 mutant (SIN3B^{PHF12D57A}) complex, and the H3K27 deacetylation experiment is also per-formed with the HDAC2 apo enzyme. Quantifications from western blot data (performed in triplicate and one replicate per experiment are shown in **e–g**) are plotted in (**h**). Data are presented as mean values ± S.D. of independent experiments, *n* = 3.

in apo form or a SIN3B complex carrying the SIN3B^{PHF12D57A} point mutant (Fig. 5d) which is unable to bind the histone H3 tail (Fig. 5f, h). This demonstrates that the SIN3B complex prefers deacetylating H3K27 over H3K9 and that H3K27 deacetylation depends on the SIN3B HDAC holo-complex assembly and specifically on the PHD finger 1 of PHF12 within the substrate receptor of SIN3B as shown in our structure.

## Discussion

Here we report the high-resolution structure of a full-length human HDAC complex which includes catalytic site, corepressor HDAC activating subunit, and histone recognition module. Our structure unveils how a SIN3-family HDAC complex assembles. We find an unexpected mechanism of activation for the drug target HDAC1/2 by the corepressor subunit SIN3B, which contacts the HDAC allosteric basic patch directly and completely encircles all the HDAC cation binding sites. Future endeavours are needed to determine the exact structural basis of SIN3A complex regulation by InsP[59].

By providing the structure of SIN3B complex with a substrate mimic, we also show structural evidence that the corepressor SIN3B establishes interactions with the histone substrate for ensuring specific deacetylation. This is the first reported example to our knowledge where a corepressor protein within an HDAC complex regulates the specific HDAC activity by contacting the substrate directly. Con-versely, it has been reported the presence of an auto-inhibitory loop in regulating the activity of the p300 acetyltransferase (HAT)[60]. Intrigu-ingly, the SIN3B^{Gate loop} is phosphorylated (Supplementary Data 1) and ubiquitylated[61], in the future it would be interesting to investigate if these post-translational modifications could influence the HDAC activity of SIN3B similarly as SUMOylation influences the activity of p300-related HATs[62].

It will also be important investigating the function of RbBP7, an additional SIN3B-associated protein[39] which we included in our SIN3B preparations for allowing solubility of SIN3B. RbBP7 is unresolved in our cryo-EM maps. An explanation for the latter could be that RbBP7 is flexibly tethered to the SIN3B complex. This is supported by our XL-MS data (Supplementary Fig. 7c) showing that RbBP7 mainly crosslinks with disordered regions within our structures.

Moreover, a model of a histone tail bound to SIN3B is consistent with histone binding through the PHD finger of SIN3B at a specific distance from the active site. This dictates which lysine residues can be fed in the catalytic tunnel, showing a specificity for H3K27ac over H3K9ac. A preference for H3K27 over H3K9 deacetylation by SIN3B was also observed in a previous study[63].

Furthermore, the interaction between SIN3B^{PAH3} and PHF12 that we describe is important because the PAH domains of SIN3 proteins are drug targets[64,65].

These findings pave the way for a model of specificity of HDAC complexes, which is crucial in understanding their general role in gene expression and chromatin structure regulation. Finally, our structure provides a resource of protein-protein interactions of an HDAC complex, which rationalises an ensemble of previous cellular and biochemical data and will guide the design of more specific and effective anti-cancer drugs.

## Methods

### Protein expression and purification

Codon optimised cDNAs of the SIN3B complex (C-terminally StrepII-tagged SIN3B isoform 2 wild type, E436A/D437A and E456R/D457R/E461R mutants), N-terminally GST-tagged MORF4L1, PHF12 (full-length, residues 1-502 and D57A mutant), RbBP7, and HDAC2 were purchased from GeneArt (Thermo Fischer Scientific) or GenScript and cloned into MultiBac vectors (pFastBac1 or pACEBac1) for insect cell expression.

Baculoviruses of each subunit were produced in Sf9 cells then co-infected in High Five cells at a density of $1.5 \times 10^6$ cells/ml using Sf-900 III medium supplemented with penicillin/streptomycin. Cells were shaken at 27 °C for 48 h then harvested and stored at −80 °C until processing.

Cell pellets were resuspended in cold lysis buffer (50 mM HEPES pH8, 500 mM NaCl, 0.5 mM TCEP, 5% glycerol, 1 mM EDTA, 0.5 mM PMSF, 2 mM benzamidine, 5 U/ml benzonase (Sigma), cOmpleteTM mini EDTA-free protease inhibitors (Roche)), lysed by sonication, and centrifuged. The supernatant was filtered and loaded on a Strep-Tactin Superflow (30060, Qiagen) column then washed with 10 CV 500 mM NaCl wash buffer (containing 500 mM NaCl, 50 mM HEPES pH 8, 5% glycerol, 1 mM EDTA), followed by 4 CV 1 M NaCl wash buffer, followed by 10 CV 200 mM NaCl wash buffer before elution using 200 mM NaCl wash buffer supple-mented with 2.5 mM d-Desthiobiotin. The eluted fractions were subsequently loaded onto a GSTrap^{TM} HP column preequilibrated with 200 mM NaCl wash buffer, followed by elution with 10 mM reduced glutathione in 200 mM NaCl. Samples were concentrated and the StrepII and GST tags were removed by O/N incubation with 3C protease at 4 °C. SEC was used for final polishing on a Superose 6 Increase 10/300 GL column (Cytiva) in buffer con-taining 20 mM HEPES pH 8, 150 mM NaCl, and 0.5 mM TCEP. Peak fractions were pooled, concentrated up to 2 µM, and used for further experiments or flash frozen in liquid nitrogen and stored at −80 °C.

In order to express the four-subunits SIN3B complex also either RbBP4 or RbBP7 must have been included in the co-expression. Although this protein is not visualised in any of our cryo-EM recon-structions, presumably because of its flexible tethering to the SIN3B complex, it allowed solubility of SIN3B. We also did not find any dif-ference in the cryo-EM reconstructions between the SIN3B^{FL} complex and the SIN3B complex where we deleted the C-terminus of PHF12 (i.e. PHF12^{ΔC-terminus}) thereby supporting the idea that also this portion of PHF12 is disordered.

### NCP167 assembly

The unmodified NCP167 and the NCP167 with the H3K9/K14/K27ac modifications were purchased from ActiveMotif.

### Histone deacetylation activity assay using the Boc-Lys(Ac)-AMC substrate

For HDAC assays using the fluorogenic substrate, Boc-Lys(Ac)-AMC[66], 100 nM protein was incubated with 100 μM Boc-Lys(ac)-AMC substrate (Cayman Chemical) in the presence or absence of 100 μM InsP4 in pH 7.4 TBS buffer for 1 h. The reaction was stopped using 200 μM SAHA inhibitor for 5 min and deacetylated substrates were digested with 50 mg/ml trypsin for 1 h. Fluorescence was measured using a POLARstar Omega (BMG Labtech) plate reader at 355/460 nm. Incubation steps were carried out at 37 °C with agitation. Unpaired $t$-tests were applied to detect differences between samples and $p$ values <0.05 was considered as statistically significant.

### Histone deacetylation activity assay using acetylated nucleosomes

In total, 100 nM H3K9ac or H3K27ac nucleosomes (Active Motif) were incubated on ice with 100 nM protein in buffer containing 25 mM HEPES pH 7.5, 50 mM NaCl, and 0.1 mM TCEP. Reactions were stopped at various timepoints by quenching with gel loading buffer (NuPAGE LDS sample buffer, Life Technologies), loaded onto 4–12% SDS-PAGE gels and transferred onto PVDF membranes for western blotting using site-specific primary antibodies, H3K9ac, H3K27ac, H3K14ac and H4 (Cell Signalling #9649, #D5E4, Sigma-Aldrich #07-353, and Abcam #ab7311 respectively). Membranes were subsequently probed with goat-anti-rabbit-HRP conjugated secondary antibody (Abcam, #ab205718) then incubated with ECL for detection and quantified using ImageJ Fiji v2.1.0/1.53c. All the primary antibodies were diluted 1:2000. The secondary antibody was diluted 1:10,000.

### Gradient fixation (GraFix)

A continuous gradient containing 20 mM HEPES pH 7.5, 150 mM NaCl, 0.5 mM TCEP, 10–30% sucrose, 0–0.1% glutaraldehyde was generated using Gradient Master (Biocomp). In total, 100 μl of SIN3B complex at the concentration of 1.7 μM was applied on top of the gradient, and centrifuged at 165,100 × $g$ for 16 h at 4 °C using SW 60 Ti ultracentrifuge. Samples were manually fractionated by hand, and the crosslinking was quenched by adding Tris-HCl pH 7.5 to a final concentration of 35 mM. GraFix[67] fractions were analysed on SDS-PAGE, and fractions that contain protein band close to the size of SIN3B complex were pooled, and buffer exchanged to 20 mM Tris-HCl (pH 8.0), 50 mM NaCl, 0.5 mM TCEP. For SIN3B:SAHA complex, SIN3B was mixed with 50 μM SAHA, and incubated for 30 min at 4 °C prior to GraFix. GraFix solution and final buffer exchange solution were supplemented with 50 μM SAHA.

### Cryo-EM grid preparation

Quantifoil R1.2/1.3 Cu 300 grids were previously glow discharged using Easiglow (Pelco) at 15 mA for 1 min. Two μl of 1 μM GraFix-treated SIN3B/SIN3B:SAHA was applied to grids, and blotted for 5 s before being plunged into liquid ethane. Grids were prepared at 4 °C and 100% humidity using a Vitrobot mark IV (Thermo Fisher).

### Cryo-EM data collection and processing

All data were collected using EPU 3 software (Thermo Fisher).

**SIN3B^FL and SIN3B complex.** We collected 25,379 EER images of the SIN3B^FL sample at ICR on a Glacios Cryo-TEM 200 kV equipped with a Falcon4i operated in counting mode at a pixel size of 0.567 Å per pixel. We collected 48,837 movie stacks in MRC format of the SIN3B sample on a Titan KRIOS 300 kV cryo-TEM at eBIC (Diamond light source, BI21809-37 session), equipped with a K3 detector operated in super-resolution bin 2x mode, at a nominal magnification of 165 kx, which yielded a pixel size of 0.513 Å per pixel. These movies were live processed during collection using the cryoSPARC live programme[68].

Movie stacks were frame aligned and binned four times. Images with resolution better than 6 Å and a total motion less than 30 pixels (estimated during frame alignment) were selected for further processing. A blob picker was used for initial particle picking during the live processing, and for obtaining initial 2D class averages. Templates generated from the latter, were used for template picking and TOPAZ[69] training and picking with cryoSPARC. Selected particles from 2D classifications performed with particles picked with the different methods explained above were pooled together and duplicated particles were removed by using the "remove duplicates" function implemented in cryoSPARC. This step removed doubly picked particles within 50 Å (shortest dimension of the SIN3B particles is ~50 Å). These particles were piped into the "ab initio model" function implemented in cryoSPARC. These particles were then imported in RELION-3.1.1[70] as described in[71] Refinement, Bayesian polishing was performed as in ref. [71] and as illustrated in the Supplementary Figs. 2, 3 and 4. Further SIN3B^core map improvement was performed by performing 3D classification without alignment with $T = 30$, with a soft mask on the SIN3B^core subcomplex, and by performing CTF refinement in RELION. This yielded a map at the resolution of 2.8 Å. The SIN3B complex map was obtained by an initial overall 3D classification with angular sampling of 7.5 degrees, followed by 3D classification focusing on the substrate recognition module including (SIN3B^PHD1:MORF4L1^MRG:PHF^SID1 subcomplex). Conversely, the final SIN3B^FL complex map was obtained by using a mask including the entire SIN3B complex (Supplementary Fig. 2) during the latest 3D classification steps.

**SIN3B:SAHA complex.** We collected 29,661 Movie stacks in MRC format on a Titan KRIOS 300 kV cryo-TEM at eBIC (Diamond light source, BI21809-41 session), equipped with a K3 detector operated in super-resolution bin 2x mode, at a nominal magnification of 165 kx, which yielded a pixel size of 0.508 Å per pixel. Processing of these data was performed similarly to the SIN3B^core complex except that two cycles of Bayesian polishing followed by 3D classification were performed. The first 3D classification was performed at $T = 20$, the second at $T = 40$.

### Model building

The PDB ID: 6XEC, was used as initial template for building HDAC2 in the SIN3B-apo structure in Coot[72]. SIN3B was build de novo based on the excellent quality of our cryo-EM map, AlphaFold[73] models for the individual SIN3B domains were used as initial templates. The PDB ID: 2LKM, was used as an initial template to model the MORF4L1^MRG:PHF12^SID1 subcomplex. PHF12 NTH, PHD1 and PHD2 were built de novo based on the excellent quality of our cryo-EM map, AlphaFold models for the individual domains were used as initial templates. Structural model refinement was performed with PHENIX real-space refinement[74] at the resolution of 2.8 Å for the SIN3B^core complexes and at either 3.4 or 3.7 Å for the SIN3B^FL and the SIN3B complex respectively. For building the SIN3B:SAHA structure, the structure of SIN3B was used as an initial template, and the SAHA ligand was fitted unambiguously.

### Crosslinking mass spectrometry analysis

In total, 30 μl of freshly purified SIN3B complex at a concentration of 1.7 μM was incubated with 0.75 μl DSSO (disuccinimidyl sulfoxide) (A33545, Thermo Scientific) pre-dissolved in DMSO at a concentration of 100 mM. Sample was incubated on ice for 10 min followed by quenching with 2 μl 500 mM Tris pH8. The sample was analysed by mass spectrometry as follows.

After the crosslinking reaction, triethylammonium bicarbonate buffer (TEAB) was added to the sample at a final concentration of 100 mM. Proteins were reduced and alkylated with 5 mM tris-2-carboxyethyl phosphine (TCEP) and 10 mM iodoacetamide (IAA)

simultaneously for 60 min in dark and were digested overnight with trypsin at final concentration 50 ng/µl (Pierce). Sample was dried and peptides were fractionated with high-pH Reversed-Phase (RP) chromatography using the XBridge C18 column (2.1 × 150 mm, 3.5 µm, Waters) on a Dionex UltiMate 3000 HPLC system. Mobile phase A was 0.1% v/v ammonium hydroxide and mobile phase B was acetonitrile, 0.1% v/v ammonium hydroxide. The peptides were fractionated at 0.2 ml/min with the following gradient: 5 min at 5% B, up to 12% B in 3 min, for 32 min gradient to 35% B, gradient to 80% B in 5 min, isocratic for 5 min and re-equilibration to 5% B. Fractions were collected every 42 s, SpeedVac dried and orthogonally pooled into 12 samples for MS analysis.

LC-MS analysis was performed on the Dionex UltiMate 3000 UHPLC system coupled with the Orbitrap Lumos Mass Spectrometer (Thermo Scientific). Each peptide fraction was reconstituted in 30 µl 0.1% formic acid and 15 µl were loaded to the Acclaim PepMap 100, 100 µm × 2 cm C18, 5 µm trapping column at 10 µl/min flow rate of 0.1% formic acid loading buffer. Peptides were then subjected to a gradient elution on the Acclaim PepMap (75 µm × 50 cm, 2 µm, 100 Å) C18 capillary column connected to a stainless steel emitter with integrated liquid junction (cat# PSSELJ, MSWIL) on the EASY-Spray source at 45 °C. Mobile phase A was 0.1% formic acid and mobile phase B was 80% acetonitrile, 0.1% formic acid. The gradient separation method at flow rate 300 nl/min was as follows: for 95 min gradient from 5–38% B, for 5 min up to 95% B, for 5 min isocratic at 95% B, re-equilibration to 5% B in 5 min, for 10 min isocratic at 5% B. Precursors between 375–1600 $m/z$ and charge equal or higher than +3 were selected at 120,000 resolution in the top speed mode in 3 s and were isolated for stepped HCD fragmentation (collision energies % = 25, 30, 36) with quadrupole isolation width 1.6 Th, Orbitrap detection with 30,000 resolution and 54 ms Maximum Injection Time. Targeted MS precursors were dynamically excluded for further isolation and activation for 30 s with 10 ppm mass tolerance.

Identification of crosslinked peptides was performed in Proteome Discoverer 2.4 (Thermo) with the Xlinkx search engine in the MS2 mode for DSSO/+158.004 Da (K). Precursor and fragment mass tolerances were 10 ppm and 0.02 Da respectively with maximum 2 trypsin missed cleavages allowed. Carbamidomethyl at C was selected as static modification.

Linear peptides were identified using the SequestHT node with precursor and fragment mass tolerances at 20 ppm and 0.02 Da respectively. Dynamic modifications included Carbamidomethyl at C, Deamidation of N/Q, DSSO Hydrolyzed (+176.014 Da) and DSSO Tris (+279.078 Da) at K, as well as Phosphorylation of S, T and Y. Spectra were searched against a UniProt FASTA file containing 21,154 Trichoplusia ni (7111) entries concatenated with the sequences of the five proteins in the complex using the O75182-2 isoform of SIN3B. Crosslinked and linear peptides were filtered at FDR < 0.01 using the Percolator node and target-decoy database search. Phosphorylation localisation probabilities were estimated with the IMP-ptmRS node.

### Reporting summary
Further information on research design is available in the Nature Portfolio Reporting Summary linked to this article.

## Data availability
The structural coordinates have been deposited in the Protein Data Bank with the following accession numbers: PDB-ID 8C60, 8BPA, 8BPB and 8BPC; the EM data have been deposited in the Electron Microscopy Data Bank with the following accession numbers: EMD-16449, EMD-16147, EMD-16148 and EMD-16149. The mass spectrometry proteomics data have been deposited to the ProteomeXchange Consortium via the PRIDE[75] partner repository with the dataset identifier PXD040006 [https://www.ebi.ac.uk/pride/login]. All the other data and materials used for the analysis are available from the corresponding author upon reasonable request. Source data are provided with this paper.

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

## Acknowledgements

We thank Chris Richardson for the IT support. We thank Fabienne Beuron for her support within the ICR EM facility. Stephen Hearnshaw for his help within the ICR biophysics facility. We thank Ruth Knight for helping with insect cell culture. We thank Alex Radzisheuskaya, Basil J. Greber and Vlad Pena for critically reading and commenting on this manuscript. We thank David Barford for discussions and for allowing CA to start this project in his laboratory with a Brenner postdoc price awarded to C.A. (Max Peruz Fund, Laboratory of Molecular Biology, Cambridge 2018). We thank Diamond for access and support of the cryo-EM facilities at the UK national electron Bio-Imaging Centre (eBIC), proposal BI21809-34, funded by the Wellcome Trust, MRC and BBSRC. M.S.M.W., R.M., M.G.K. and C.A. are supported by the Sir Henry Dale Fellowship 215458/Z/19/Z. M.S.M.W. and R.M. are also supported by the Institute of Cancer Research (ICR), grant number allocated are GFR005X and GFR146X, respectively. The work of T.I.R. and J.S.C. was funded by the CRUK Centre grant with reference number C309/A25144.

## Author contributions

M.S.M.W. cloned and reconstituted all the SIN3B constructs in this study. M.S.M.W. performed all the biochemical assays in this study. R.M. performed GraFIX, prepared cryo-EM grids and helped C.A. to screen them. C.A. coordinated the EM pipeline, screened the cryo-EM grids, collected cryo-EM data, and processed the cryo-EM data. C.A. performed structural model interpretation and building with inputs from M.G.K. M.G.K. performed atomic coordinate model refinement. T.I.R. and J.S.C. performed and supervised the MS analysis of the SIN3B apo sample respectively. C.A. directed the project and designed experiments with M.S.M.W. and R.M. C.A. prepared the manuscript and wrote the manuscript with the help of M.S.M.W. and R.M.

## Competing interests

The authors declare no competing interests.
