## [Peer Review File · Nature Communications]

Mechanism of assembly, activation and lysine selection by the SIN3B histone deacetylase complexREVIEWER COMMENTS

Reviewer #1 (Remarks to the Author):

In this study, Wan et al. report the cryo-EM structure of the chromatin-associated human Sin3B-HDAC holocomplex. A homolog of the yeast Rpd3S complex. Using the purified recombinant proteins Sin3B, HDAC2, PHF12 and MORFL1, the authors reconstitute a stable and pure Sin3B complex, which they subject to cryoEM for structure determination. They obtain a high resolution of the complex at 3.4Å resolution, as well as a core complex resolution of 2.8Å. Their analysis highlights a novel domain within Sin3B (the gate loop domain) which insets into the catalytic site of HDAC2, as well as extensive contacts between Sin3B and HDAC2. Using SAHA as a substrate mimic, they further demonstrate that the interactions between discrete domains of Sin3B and HDAC2 stabilize the substrate and stimulates HDAC activity. Furthermore, their analysis provides a molecular basis for the absence of a SANT domain protein in the Sin3B-HDAC complex, and the mutually exclusive presence of SDS3 or PHF12 in the SIN3-HDAC complexes. Finally, the structural analysis of the complex suggests a substrate specificity for H3K27 deacetylation rather than the more N terminal modification H3K9Ac. Of note, this study also reveals or confirms the presence of multiple specific chromatin binding domains within the complex.

Overall, this study yields important insight on the structure and function of the understudied Sin3B-HDAC complex. The analysis conducted here not only justifies some previous unexplained findings (the absence of a SANT domain within the Sin3B complex, the exclusive binding to PHF12 rather than SDS3, and the substrate preference of the complex), it also unveils a novel molecular mechanisms for the activation of a chromatin modifying complex through its interaction with scaffolding subunits. Importantly, the elegant enzymatic and biochemical assays used for crossvalidation bring confidence in the interpretation of the structural data.

A few points should be addressed.

- Do the authors expect InsP4 addition to stimulate HDAC activity in the absence of Sin3B? (contrary to what is shown on Fig. 3A).
- Given the unique role for Sin3B in this complex, could the authors point to structural difference with the highly related SIN3A protein that would justify this specificity?
- Does HDAC still form a stable complex with Sin3B mutated for Glu436 and Asp437?
- The sentence found on lines 211-213 ("Importantly, the same Gate loop mutant..." is not clear.
- How do these findings relate to the substrate specificity identified for the Sin3B complex on specific modified histones in Wang et al, eLife 2020?
- Line 275: the authors refer to an interaction study identifying PHF12 (ref 51), but this study only investigated the interaction between PHF12 and SIN3A, not SIN3B.
- The authors mention the strong phenotypes elicited upon PHF12 genetic inactivation on line 283. However, the phenotypes in SIN3B KO cells are much milder: How do they explain this difference based on their observations?

Reviewer #2 (Remarks to the Author):

Alfieri and coworkers present the cryo-EM structure of the human SIN3B histone deacetylase holocomplex with and without a SAHA inhibitor. They also complement their studies with extensive cross-linking mass-spec and additional mutagenesis and activity data. Surprisingly, the complex reveals that SIN3B encircles most of the catalytic HDAC subunit with the PHF12 and MORF4L1 also participating in mediating HDAC interactions. Very interestingly, SIN3B contacts a basic patch, typically occupied by an IP6 in related complexes and also inserts a loop into a catalytic tunnel, together contributing to HDAC catalytic activity. The data presented also supports a model where regions of PHF12 and MORF4L1 contribute to histone binding.

Taken together, the study is of significant interest to the field as it illustrates how HDAC complex subunits can contribute to substrate binding and catalysis and also has potential implications for the rational development of more selective class I HDAC inhibitors. The manuscript is also generally clearly presented and illustrated and the studies quite rigorous. Nonetheless, there are several areas that need to be addressed before publication.

1. Overall, the introduction is quite short and leaves out many of concepts that are required to understand the results section. For example, roles of calcium binding sites and a basic patch within class I HDACs, and the relationship of the current structure to the HDAC complexes NuRD, MiDAC and SMART need to be introduced in the introduction section. In addition, there are many references to multiple figure panels, with the one key panel for supporting the sentence often buried. Can the authors please restrict their references, ideally to a single figure panel, or at most 2 figure panels.

2. The first sentence of the abstract is misleading: "Histone deacetylase complexes remove histone lysine acetylation, a key post-translational modification that activates transcription at each gene." This could be misinterpreted to mean HDACs mediate gene activation, which is not true.

3. Word choice in several sentences in the introduction should be reconsidered:

Line 49 reads "biochemical and structural information on the full-length version of these complexes," but should read "biochemical and structural information on the intact version of these complexes"

4. RBBP7 is expressed and purified with the SIN3B complex, but no mention of the protein is made in the main text (stoichiometry, function, reason for being present but unresolved, etc.). The role of the protein in SIN3B complex function should be mentioned, and the lack of ability to resolve it should be mentioned (not just in the supplement). The reasoning given in the supplement for failing to resolve RBBP7 is flexibility relative to SIN3B, and the 2D Classes show smeared density which may indicate a flexible region as proposed, but the scale of the smearing is inconsistent between data sets. Have the authors tried data processing pipelines focused on resolving the RBBP7/smeared density?

5. No cross-links to RBBP7 are mentioned in the XL-MS data. Why are they not mentioned/not found?

6. The masked and phase randomized FSC curves for the cryo-EM maps need to be reported.

7. The cryo-EM maps need orientation distributions for the final maps (bild file, azimuth plot, etc.) and a sphericity score from 3DFSC or Eod score from cryoEF

8. Main text figures should contain the cryo-EM maps for readers to assess.

9. Map-to-model FSC curves should be reported

10. Line 50 reads "are challenging to reconstitute and structurally analyse for their high intrinsic disorder," but should read "are challenging to reconstitute and structurally analyze because of their high intrinsic disorder"

11. Line 51 reads "In mammals, two highly similar proteins named SIN3A and SIN3B, take part in," should read "In mammals, two highly similar proteins named SIN3A and SIN3B 14, are part of"

12. Line 115, if a13 is numbered and labeled, then the referenced b-strands should be numbered and labeled as well.

13. Line 117, A second HDAC Ca²⁺ binding site is mentioned without first discussing the role of the calcium binding sites in class I HDACs. The Ca²⁺ binding sites need to at least be discussed in the introduction section.

14. Lines 111 and 129, The title of this section and the conclusion implies that what is discussed explains why HDAC/SIN3B interaction stimulates catalysis, but this is not the case as the discussion does not directly demonstrate how catalysis is stimulated. Stimulation of catalysis is only implicated from related structures.
15. Line 161, a24-a25 is referenced but is unlabeled in Figure 2b.
16. Line 172, the gate loop in figure 2b needs to be labeled.
17. Figure 3a and b should be with figure 2 where the structural details are shown.
18. Lines 218-222. The discussion of a potential analogy to p300 acetyltransferase regulation detracts from the main conclusion of this section and should be moved to the discussion section.
19. Line 246, The "model" for histone tail binding by PHF12 is treated too much like an actual structure with a bound histone tail. I would suggest changing the language from "indicates that PHF12 is indeed able to present the histone H3 tail to the catalytic tunnel of the HDAC starting from Lys14" to "indicates that PHF12 could present the histone H3 tail to the catalytic tunnel of the HDAC starting from Lys14"
20. Lines 257-259, Related to the point above, I think the statement about how the histone is delivered to the active site is made too strongly since it is based on a model of histone binding. I would suggest changing "establishes" to "suggests"
21. Line 281 should read "likely constitutes a substrate receptor by recruiting the H3 histone tail with its PHD1"
22. It might be easier to follow to have the assays in Figure 5 should also accompany the corresponding structural figures
23. Line 308, what do the authors mean by "SIN3B establishes transient interactions with the histone substrate." I may have missed the evidence that the interaction is transient.
24. Line 309, the authors state "we show that the histone recognition module of SIN3B recruits the histone tail via a PHD finger," but this is based only on a model so the authors should say something to the effect "a model of a histone tail bound to SIN3B is consistent with histone binding through the PHD finger of SIN3B"

Reviewer #3 (Remarks to the Author):

The paper presented the first structure of SIN3B/HDAX2 with subunit PHF12 and MORF4L1 in 1:1:1:1 ratio through recombinant SIN3B complex. Using the cryo-EM structure and XL-MS the authors indicated a SIN3B loop inserted into the catalytic tunnel to stabilize deacetylation substrate, a SIN3B MD contacted a second Calcium binding site of HDAC2, a SIN3B HID directly contacted HDAC2 basic patch that could compete with InsP co-factor, a PHF12 scaffolded the complex. The research is critically important for both basic medical research and clinical drug discovery. The paper clearly evidenced what the authors claimed in most part except the following parts may need to be addressed:

1. Supplementary Table 2

Missed the distributions of XL lysine Ca-Ca distance mapped to the cryo-EM structure and how much percentages out of the DSSO linkage limits. This is the most direct way to test if XL MS results agree

with cryo-EM structure. Please justify it if big portion of XL-MS distances are out of the DSSO linkage limits (35 Å).

2. Supplementary Information

Materials and Methods

Crosslinking mass spectrometry analysis

1) DSSO is a CID cleavable crosslinker, XL MS data can be acquired as MS2 or MS2MS3. Usually, more flexible database searching strategies and FDR control can be realized through MS2MS3 workflow. Please justify why MS2 instead of MS2MS3 was used for DSSO XL-MS.

2) Since all recombinant human proteins were expressed in baculovirus/insect cell system, XLMS data set should be searched against host cell proteins sequences plus human SIN3 subunit sequences. However, authors searched XL MS data set against human protein sequence database. The whole data set need to be re-searched against correct database.

3) As authors mentioned that phosphorylation and ubiquitination are important PTMs for SIN3B complex, XLMS data could be searched including these two modifications.

3. The SIN3B HID directly contacting HDAC2 basic patch is interesting in the recombinant SIN3B complex. Is it possible in vivo the other subunit(s) will join the complex and change the interactions between SIN3B HID and HDAC2 in a way that InsP can act as an effective co-factor? It might be worthwhile to justify this possibility in the paper.

8th March, 2023.

Re: Manuscript NCOMMS-23-03692-T

Reviewer #1 (Remarks to the Author):

In this study, Wan et al. report the cryo-EM structure of the chromatin-associated human Sin3B-HDAC holocomplex. A homolog of the yeast Rpd3S complex. Using the purified recombinant proteins Sin3B, HDAC2, PHF12 and MORFL1, the authors reconstitute a stable and pure Sin3B complex, which they subject to cryoEM for structure determination. They obtain a high resolution of the complex at 3.4Å resolution, as well as a core complex resolution of 2.8Å. Their analysis highlights a novel domain within Sin3B (the gate loop domain) which insets into the catalytic site of HDAC2, as well as extensive contacts between Sin3B and HDAC2. Using SAHA as a substrate mimic, they further demonstrate that the interactions between discrete domains of Sin3B and HDAC2 stabilize the substrate and stimulates HDAC activity. Furthermore, their analysis provides a molecular basis for the absence of a SANT domain protein in the Sin3B-HDAC complex, and the mutually exclusive presence of SDS3 or PHF12 in the SIN3-HDAC complexes. Finally, the structural analysis of the complex suggests a substrate specificity for H3K27 deacetylation rather than the more N terminal modification H3K9Ac. Of note, this study also reveals or confirms the presence of multiple specific chromatin binding domains within the complex.

Overall, this study yields important insight on the structure and function of the understudied Sin3B-HDAC complex. The analysis conducted here not only justifies some previous unexplained findings (the absence of a SANT domain within the Sin3B complex, the exclusive binding to PHF12 rather than SDS3, and the substrate preference of the complex), it also unveils a novel molecular mechanisms for the activation of a chromatin modifying complex through its interaction with scaffolding subunits. Importantly, the elegant enzymatic and biochemical assays used for crossvalidation bring confidence in the interpretation of the structural data.

A few points should be addressed.

1. Do the authors expect InsP4 addition to stimulate HDAC activity in the absence of Sin3B? (contrary to what is shown on Fig. 3A).

We acknowledge the reviewer for giving us the opportunity to better describe our results in Fig. 3A (now changed to Fig. 2d because of reviewer 2 point 17). We do not expect that InsP4 would stimulate the activity of the HDAC enzyme in isolation. This because the InsP4-dependant stimulation requires a SANT domain from a corepressor binding partner as demonstrated in PMID: 23791785. To clarify this, we added the following sentence and reference PMID: 23791785 into our main text: "Consistently with [PMID: 23791785] InsP4 does not stimulate the activity of the HDAC apo enzyme in the absence of a SANT domain containing corepressor binding partner." [Lines 206-207, page 5].

2. Given the unique role for Sin3B in this complex, could the authors point to structural difference with the highly related SIN3A protein that would justify this specificity?

SIN3A and SIN3B paralogs are ~70% similar, and we notice that high similarity is also present in the interaction surfaces engaged with SIN3B-specific components that we describe here. For this reason, in absence of a high-resolution structure of the SIN3A complex, it is not yet possible to explain how SIN3A and SIN3B determine the assembly of two distinct complexes each with their specific subunits. To clarify this, we added the following sentence in our main text: “Nevertheless, to fully understand the specificity mechanisms that cause the assembly of SIN3A and SIN3B proteins into two different complexes, requires future endeavours in determining the high-resolution structure of the SIN3A complex.” [Lines 416-418, page 9].

3. Does HDAC still form a stable complex with Sin3B mutated for Glu436 and Asp437?

Yes, the expressed and purified SIN3B^{E436A/D437A} mutant is stable and stoichiometric as the wild-type SIN3B complex. We added one SDS page showing the quality of this protein complex preparation in Supplementary Figure 1c.

4. The sentence found on lines 211-213 (“Importantly, the same Gate loop mutant...” is not clear.

We acknowledge the reviewer for giving us the chance to elaborate on this sentence in the main text, now changed to: “Notably, although the SIN3BE436A/D437A mutant is no longer able to deacetylate the specific H3K27 from a nucleosome, it still manifests HDAC activity with the fluorogenic acetyl-lysine substrate (Fig. 3g). These data together support the idea that the SIN3B Gate loop allows for deacetylation of the specific target substrate (i.e. the acetylated nucleosome), where the lysine residues within the long histone tail need to be optimally presented into the SIN3B complex active site. Conversely, the fluorogenic substrate used in our HDAC assay, given its small size, could have easier access to the SIN3B complex active site, where it can still be processed in the SIN3BE436A/D437A mutant.” [Lines 294-301, page 7, also lines 113-114, page 3].

5. How do these findings relate to the substrate specificity identified for the Sin3B complex on specific modified histones in Wang et al, eLife 2020?

This is a very good point raised by the reviewer. We added the following comment with the Wang et al. reference in our main text: “A preference for H3K27 over H3K9 deacetylation by SIN3B was also observed in a previous study [PMID: 32501215].” [492-493, page 11].

The overall low activity observed in the SIN3B complex from Wang et al. could be explained by the absence of two subunits in the expression system within this study (i.e. PHF12 and MORF4L1). We find that these two subunits are essential for the formation of the histone recognition module of the SIN3B complex and for the nucleosome directed deacetylation by SIN3B (please see Figure 5 and the effect of the PHF12^{PHF12D57A} mutant, Figure 5f). Substrate preferences of the SIN3B complex in Wang et al. could be due to small amounts of endogenous PHF12/MORF4L1

protein co-purified with the transiently expressed subunits SIN3B, HDAC and RBBP4.

6. Line 275: the authors refer to an interaction study identifying PHF12 (ref 51), but this study only investigated the interaction between PHF12 and SIN3A, not SIN3B. We acknowledge the reviewer for spotting this inconsistency, we removed ref 51 from the main text. [Line 408, page 9].
7. The authors mention the strong phenotypes elicited upon PHF12 genetic inactivation on line 283. However, the phenotypes in SIN3B KO cells are much milder: How do they explain this difference based on their observations? The SIN3B complex is crucial for organismal development as it plays a role in several developmental pathways. [Line 46, page 2]. This could explain the strong phenotype of the PHF12 mouse KO. In adult cultured cells, SIN3B has been found to be a regulator of cell cycle dependent transcription and a promoter of cellular quiescence/senescence. SIN3B depletion causes the loss of ability to exit the cell cycle (G0) but cells can still proliferate [PMID: 19654306, PMID: 18332431 and PMID: 30517867]. Our data shows that PHF12 is crucial for the SIN3B biochemical function and given that the SIN3B pathway is essential in development, and less important for cell proliferation in adult cells, we expect that the strongest phenotype of SIN3B pathway inactivation would be during development.

Reviewer #2 (Remarks to the Author):

Alfieri and coworkers present the cryo-EM structure of the human SIN3B histone deacetylase holo-complex with and without a SAHA inhibitor. They also complement their studies with extensive cross-linking mass-spec and additional mutagenesis and activity data. Surprisingly, the complex reveals that SIN3B encircles most of the catalytic HDAC subunit with the PHF12 and MORF4L1 also participating in mediating HDAC interactions. Very interestingly, SIN3B contacts a basic patch, typically occupied by an IP6 in related complexes and also inserts a loop into a catalytic tunnel, together contributing to HDAC catalytic activity. The data presented also supports a model where regions of PHF12 and MORF4L1 contribute to histone binding.

Taken together, the study is of significant interest to the field as it illustrates how HDAC complex subunits can contribute to substrate binding and catalysis and also has potential implications for the rational development of more selective class I HDAC inhibitors. The manuscript is also generally clearly presented and illustrated and the studies quite rigorous. Nonetheless, there are several areas that need to be addressed before publication.

1. Overall, the introduction is quite short and leaves out many of concepts that are required to understand the results section. For example, roles of calcium binding sites and a basic patch within class I HDACs, and the relationship of the current structure to the HDAC complexes NuRD, MiDAC and SMART need to be introduced in the introduction section. In addition, there are many references to multiple figure panels, with the one key panel for supporting the sentence often

buried. Can the authors please restrict their references, ideally to a single figure panel, or at most 2 figure panels.

We thank the reviewer for this comment. We now extended the introduction with the details requested (i.e. roles of calcium binding sites and introduction about HDAC complexes). [Lines 43-44, 50-55 and 62-68, page 2].

We also limited referencing to 2 figure panels throughout the text.

2. The first sentence of the abstract is misleading: “Histone deacetylase complexes remove histone lysine acetylation, a key post-translational modification that activates transcription at each gene.” This could be misinterpreted to mean HDACs mediate gene activation, which is not true.

We thank the reviewer for spotting this ambiguous sentence which has been changed to: “Lysine acetylation in histone tails is a key post-translational modification that controls transcription activation. Histone deacetylase complexes remove histone acetylation, thereby repressing transcription and regulating the transcriptional output of each gene.” [Lines 18-20 page 1].

3. Word choice in several sentences in the introduction should be reconsidered: Line 49 reads “biochemical and structural information on the full-length version of these complexes,” but should read “biochemical and structural information on the intact version of these complexes”

We incorporated the suggested word choice, thank you. [Lines 58-61, page 2].

4. RBBP7 is expressed and purified with the SIN3B complex, but no mention of the protein is made in the main text (stoichiometry, function, reason for being present but unresolved, etc.). The role of the protein in SIN3B complex function should be mentioned, and the lack of ability to resolve it should be mentioned (not just in the supplement). The reasoning given in the supplement for failing to resolve RBBP7 is flexibility relative to SIN3B, and the 2D Classes show smeared density which may indicate a flexible region as proposed, but the scale of the smearing is inconsistent between data sets. Have the authors tried data processing pipelines focused on resolving the RBBP7/smeared density?

We thank the reviewer for this comment/question. We indeed tried data processing pipelines focused on resolving the smeared density that the reviewer mentions. Unfortunately, we failed in resolving RBBP7 from it. A possible reason for this is that the small mass of RBBP7 on its own presumably bound to a flexibly tethered peptide from the SIN3B complex would be at the limit of signal/noise ratio that would allow particle alignment and success in our attempts. We now added this information in the Discussion section: “It will also be important investigating the function of RbBP7, an additional SIN3B-associated protein [PMID: 32467258] which we included in our SIN3B preparations for allowing solubility of SIN3B. RbBP7 is unresolved in our cryo-EM maps. An explanation for the latter could be that RbBP7 is flexibly tethered to the SIN3B complex. This is supported by our XL-MS data (Supplementary Figure 7c) showing that RbBP7 mainly crosslinks with disordered regions within our structures.” [Lines 469-488, pages 10-11].

5. No cross-links to RBBP7 are mentioned in the XL-MS data. Why are they not mentioned/not found?

As mentioned in the previous point, RbBP7 mainly crosslinks with disordered regions within our structures. We now added these data in Supplementary Figure 7c.

6. The masked and phase randomized FSC curves for the cryo-EM maps need to be reported.

We added these curves in Supplementary Figure 5, thank you.

7. The cryo-EM maps need orientation distributions for the final maps (bild file, azimuth plot, etc.) and a sphericity score from 3DFSC or Eod score from cryoEF

We added these data in Supplementary Figure 5, thank you.

Because the SIN3B structure at 4.1Å was at near Nyquist, initially we could not calculate the 3DFSC score (the program would fail), to overcome this we polished and rescaled to bin 2x the data and this allowed an improvement of resolution to 3.7Å, Supplementary Figure 3 and Supplementary Table 1.

8. Main text figures should contain the cryo-EM maps for readers to assess.

We added the maps into our main Figures 1, 2, 3 and 4. We believe that this improved the main figures a lot, therefore we are grateful to the reviewer for this suggestion.

9. Map-to-model FSC curves should be reported

We added these curves in Supplementary Figure 5, thank you.

10. Line 50 reads “are challenging to reconstitute and structurally analyse for their high intrinsic disorder,” but should read “are challenging to reconstitute and structurally analyse because of their high intrinsic disorder”

We incorporated the suggested word choice, thank you. [Lines 60-61, page 2].

11. Line 51 reads “In mammals, two highly similar proteins named SIN3A and SIN3B, take part in,” should read “In mammals, two highly similar proteins named SIN3A and SIN3B 14, are part of”

We incorporated the suggested word choice, thank you. [Line 69, page 2]

12. Line 115, if a13 is numbered and labeled, then the referenced b-strands should be numbered and labeled as well.

We have done this in both main text [line 155, page 4] and in Figure 2, thank you.

13. Line 117, A second HDAC Ca²⁺ binding site is mentioned without first discussing the role of the calcium binding sites in class I HDACs. The Ca²⁺ binding sites need to at least be discussed in the introduction section.

We addressed this in the point N.1, thank you.

14. Lines 111 and 129, The title of this section and the conclusion implies that what is discussed explains why HDAC/SIN3B interaction stimulates catalysis, but this is not the case as the discussion does not directly demonstrate how catalysis is stimulated. Stimulation of catalysis is only implicated from related structures.

We removed these statements from the lines mentioned by the reviewer, thank you. [Lines 150 and 184, Page 4 and 5].

15. Line 161, a24-a25 is referenced but is unlabeled in Figure 2b.

We have done this, thank you. [Figure 2c].

16. Line 172, the gate loop in figure 2b needs to be labeled.

We have done this, thank you. [Figure 2c].

17. Figure 3a and b should be with figure 2 where the structural details are shown.
We have done this, thank you. [Figure 2d and e].
18. Lines 218-222. The discussion of a potential analogy to p300 acetyltransferase regulation detracts from the main conclusion of this section and should be moved to the discussion section.
We have done this, thank you. [Lines 463-465, page 10].
19. Line 246, The “model” for histone tail binding by PHF12 is treated too much like an actual structure with a bound histone tail. I would suggest changing the language from “indicates that PHF12 is indeed able to present the histone H3 tail to the catalytic tunnel of the HDAC starting from Lys14” to “indicates that PHF12 could present the histone H3 tail to the catalytic tunnel of the HDAC starting from Lys14”
We have done this and specified in Figure 5d that the H3 tail is modelled, thank you. [Lines 437-438, page 10, and Figure 5d].
20. Lines 257-259, Related to the point above, I think the statement about how the histone is delivered to the active site is made too strongly since it is based on a model of histone binding. I would suggest changing “establishes” to “suggests”
We have done this, thank you. [Lines 376, page 8].
21. Line 281 should read “likely constitutes a substrate receptor by recruiting the H3 histone tail with its PHD1”
We have done this, thank you. [Line 411, page 9].
22. It might be easier to follow to have the assays in Figure 5 should also accompany the corresponding structural figures
We have now done this, thank you. [Figure 5e-h, also lines 424-438, pages 9-10].
23. Line 308, what do the authors mean by “SIN3B establishes transient interactions with the histone substrate.” I may have missed the evidence that the interaction is transient.
The reviewer is right, we do not have any clear evidence that the interaction between SIN3B and the substrate is transient, therefore we removed the word transient. [Line 460, Page 10].
24. Line 309, the authors state “we show that the histone recognition module of SIN3B recruits the histone tail via a PHD finger,” but this is based only on a model so the authors should say something to the effect “a model of a histone tail bound to SIN3B is consistent with histone binding through the PHD finger of SIN3B”
We have done this, thank you. [Lines 489-490, page 11].

Reviewer #3 (Remarks to the Author):

The paper presented the first structure of SIN3B/HDAX2 with subunit PHF12 and MORF4L1 in 1:1:1:1 ratio through recombinant SIN3B complex. Using the cryo-EM structure and XL-MS the authors indicated a SIN3B loop inserted into the catalytic tunnel to stabilize deacetylation substrate, a SIN3B MD contacted a second Calcium binding site of HDAC2, a SIN3B HID directly contacted HDAC2 basic patch that could compete with InsP co-factor, a PHF12 scaffolded the complex. The research is critically import for both

basic medical research and clinical drug discovery. The paper clearly evidenced what the authors claimed in most part except the following parts may need to be addressed:

1. Supplementary Table 2

Missed the distributions of XL lysine Ca-Ca distance mapped to the cryo-EM structure and how much percentages out of the DSSO linkage limits. This is the most direct way to test if XL MS results agree with cryo-EM structure. Please justify it if big portion of XL-MS distances are out of the DSSO linkage limits (35 Å).

This is a very good point raised by the reviewer. We now have included in our Supplementary Table 2 (sheet named 3D distances): (i) a list of the XL lysine Ca-Ca distance mapped to the cryo-EM structure, and (ii) a histogram showing the number of crosslinks within a specific distance window. This shows that 70% (31 out of 44) of the crosslinks are below 35Å, and that 88% (23 out of 26) of crosslinks with at least 2 Crosslink Spectrum Matches (CSMs) are below 35Å. Therefore, the biggest portion of crosslinks are within the distance restrains. The crosslinks that exceed the distance restrains and feature low numbers of CSMs are the lowest abundant ones and could be due for example to low amount of partially disassembled complex.

2. Supplementary Information

Materials and Methods

Crosslinking mass spectrometry analysis

1) DSSO is a CID cleavable crosslinker, XL MS data can be acquired as MS2 or MS2MS3. Usually, more flexible database searching strategies and FDR control can be realized through MS2MS3 workflow. Please justify why MS2 instead of MS2MS3 was used for DSSO XL-MS.

Thank you for this comment. The used strategy is based on a recent study showing that the simpler and faster stepped higher-energy collisional dissociation (HCD) method outperforms the MS3-based data acquisition approaches in terms of sensitivity and specificity (PMID: 35613060).

3. Since all recombinant human proteins were expressed in baculovirus/insect cell system, XLMS data set should be searched against host cell proteins sequences plus human SIN3 subunit sequences. However, authors searched XL MS data set against human protein sequence database. The whole data set need to be re-searched against correct database.

Thank you for this comment. We have re-searched the XL MS data set against the database of the expression system (i.e. High Five insect cell, *Trichoplusia ni*) and the human SIN3B complex proteins, and updated Supplementary Table 2, Supplementary Figure 7, and Methods. [Lines 1054-1062, Page 27]. As result, we identify very similar crosslinks as in the previous analysis and 120 more crosslinks in total (~20 more with score >100). This is because changing database slightly modified the false discovery rate calculation when matching the spectra, and there are also new crosslinks on insect proteins. There is also some homology between some of the human and insect proteins (e.g probable histone-binding protein Caf1 and RBBP7). The proteins with homology that share peptides are separated with ";"

in the "Protein Descriptions" columns. For the xiView input we excluded crosslinks that contain peptides that map to *Trichoplusia ni* and are also common with the human proteins.

4. As authors mentioned that phosphorylation and ubiquitination are important PTMs for SIN3B complex, XLMS data could be searched including these two modifications.

We appreciate this very good suggestion from the reviewer. We performed phosphorylation search for linear regular peptides, and SIN3B seems quite heavily phosphorylated, including sites within and near the Gate loop. We added these data in our Supplementary Table 2 and a reference to it [line 465, Page 10]. The phosphosites per protein are shown in the sheet "Proteins-peptides" in the "modifications" column. The numbers in the parenthesis show the localization probability of the phosphate on the protein site, but this is >95% in most phosphosites. We did not search for Ubiquitination because the remnant K-GG modification (of ubiquitin after trypsin cleavage) has the same mass as lysine dicarbamidomethylation induced by iodoacetamide that we normally use for cysteine alkylation in sample preparation. Therefore, the search would return quite a lot of false positive matches in this case.

5. The SIN3B HID directly contacting HDAC2 basic patch is interesting in the recombinant SIN3B complex. Is it possible in vivo the other subunit(s) will join the complex and change the interactions between SIN3B HID and HDAC2 in a way that InsP can act as an effective co-factor? It might be worthwhile to justify this possibility in the paper.

Thank you for this comment. Our structural data suggest that InsP and SIN3B HID bind HDAC2 in a mutually exclusive manner. However, the structure of the related SIN3A complex is still unknown and perhaps the unique subunits present in the SIN3A complex could potentially allow interactions between InsP and HDAC in vivo. Evidence for the latter is shown in one study focusing on the *S. cerevisiae* SIN3A homologous complex named Rpd3(S): PMID: 31358618, we now added this reference in our main text: "Future endeavours are needed to determine the exact structural basis of SIN3A complex regulation by InsP [PMID: 31358618]." [Lines 457-458, Page 10].

REVIEWERS' COMMENTS

Reviewer #1 (Remarks to the Author):

The authors have addressed my previous concerns.

Reviewer #2 (Remarks to the Author):

The authors have done a very nice job addressing our concerns and we believe that the manuscript is now suitable for publication.

Reviewer #3 (Remarks to the Author):

The authors had addressed all of my questions, including adding Ca-Ca distance, researching data set with correct database and including modifications in data base searching. All results are impressive and can support authors conclusion.